# Zosuquidar: An Effective Molecule for Intracellular Ca^2+^ Measurement in P-gp Positive Cells

**DOI:** 10.3390/ijms25063107

**Published:** 2024-03-07

**Authors:** Livia Pelegrinova, Lucia Sofrankova, Jana Spaldova, Pavol Stefik, Zdena Sulova, Albert Breier, Katarina Elefantova

**Affiliations:** 1Institute of Molecular Physiology and Genetics, Centre of Bioscience, Slovak Academy of Sciences, Dúbravská Cesta 9, 845 05 Bratislava, Slovakia; livia.pelegrinova@savba.sk (L.P.); zdena.sulova@savba.sk (Z.S.); 2Institute of Biochemistry and Microbiology, Faculty of Chemical and Food Technology, Slovak University of Technology in Bratislava, Radlinského 9, 812 37 Bratislava, Slovakia; lucia.sofrankova@stuba.sk (L.S.); jana.spaldova@stuba.sk (J.S.); pavol.stefik@stuba.sk (P.S.)

**Keywords:** Zosuquidar, Fluo-3/AM, flow cytometry, P-glycoprotein, multidrug resistance

## Abstract

Intracellular calcium, as a second messenger, is involved in multilevel cellular regulatory pathways and plays a role (among other processes) in switching between survival and initiation of cell death in neoplastic cells. The development of multidrug resistance (MDR) in neoplastic cells is associated with the ability of cells to escape programmed cell death, in which dysregulation of intracellular calcium may play an important role. Therefore, reliable monitoring of intracellular calcium levels is necessary. However, such a role might be limited by a real obstacle since several fluorescent intracellular calcium indicators are substrates of membrane ABC drug transporters. For example, Fluo-3/AM is a substrate of P-glycoprotein (ABCB1 member of the ABC family), whose overexpression is the most frequent cause of MDR. The overexpression of ABCB1 prevents MDR cell variants from retaining this tracer in the intracellular space where it is supposed to detect calcium. The solution is to use a proper inhibitor of P-gp efflux activity to ensure the retention of the tracer inside the cells. The present study showed that Zosuquidar and Tariquidar (P-gp inhibitors) are suitable for monitoring intracellular calcium, either by flow cytometry or confocal microscopy, in cells overexpressing P-gp.

## 1. Introduction

Intracellular calcium is involved in almost all processes that ensure overall cellular homeostasis and proper cell functioning [1,2]. These processes include gene expression, post-translational modifications, chemotaxis, regulation of muscle contraction, regulation of neurotransmitter or insulin release, overall metabolism, and many others [3,4]. In certain pathological situations, there is a potential for calcium overload, resulting in increased intracellular Ca^2+^ levels. This can lead to programmed cell death or necrosis [4,5]. Therefore, sensitive measurements of intracellular calcium levels can provide valuable information.

Several calcium-sensing molecules can effectively monitor intracellular calcium levels [6,7]. Fluo-3-penta acetoxymethyl ester (Fluo-3/AM), a fluorescent calcium sensor, can serve as an example. The five acetoxymethyl (AM) groups in its molecule modify the carboxyl groups necessary for Ca^2+^ binding. Esterification of the COOH groups produces a molecule capable of penetrating the plasma membrane of the cell. In the intracellular space, AM groups are released from the ester linkage by cytoplasmic esterases to form Fluo-3, which is unable to penetrate membranes, leading to the entrapment of Fluo-3 in the cell [8,9]. In addition, free COOH groups chelate Ca^2+^ ions with high affinity. When calcium ions bind to Fluo-3, its composition changes to a conformer with typical fluorescence properties [10]. Fluorescence intensity is a measure of the level of free calcium in the cytosol and can be detected and quantified using a variety of fluorescence devices, including a fluorescence cytometer or a fluorescence/confocal microscope. The cytometric approach enables fine-scale measurements of cytoplasmic calcium equilibria. Additionally, it allows real-time monitoring and analysis of the kinetics of changes in cytosolic free Ca^2+^ levels directly in living cells [9].

To study the sensitivity of tumor-transformed cells to chemotherapeutics, established cell models derived from human or animal tumor tissues can be used. Just as resistance to chemotherapeutics occurs during the treatment of neoplastic diseases, it can also be induced in cell or tissue cultures by repeated passages in media with a sublethal concentration of chemotherapeutics. In this way, it is possible to create multidrug-resistant (MDR) variants for the specific study of cells with insufficient sensitivity to antineoplastic substances [11]. One of the most common molecular causes of MDR is the overexpression of drug transporters from the ABC protein family, such as P-glycoprotein (P-gp, ABCB1 member), Multidrug Resistance Associated Protein 1 (MPR1, ABCC1 member) or Breast Cancer Resistance Associated Protein (BCRP, ABCG2 member), and others [12,13]. Fluo-3/AM is a substrate of the transporters ABCB1 and ABCC1. ABCB1 and ABCC1 prevent its intracellular retention by decreasing its uptake and increasing its efflux [14,15,16,17]. Therefore, the transport proteins could prevent the proper loading of the fluorescent marker into living cells and could potentially cause misleading results [17,18]. However, the transport of Fluo-3/AM by the ABCG2 transporter has not been described yet.

This study aimed to determine whether two third-generation P-gp inhibitors (Zosuquidar and Tariquidar) can induce Fluo-3/AM retention in P-gp-expressing cells to perform intracellular Ca^2+^ assays. As experimental models, parental leukemia cell lines L1210 (murine lymphocytic leukemia cells), along with MOLM-13 and SKM-1 (both human acute myeloid leukemia cells), with limitingly low P-gp expression, and their P-gp positive variants, R (L1210), SKM-1/vcr, and MOLM-13/vcr, obtained by selection/adaptation via passaging in media with gradually increasing concentrations of vincristine [19], were used. In addition, a P-gp positive T (L1210) variant was also used, in which the induction of P-gp expression occurred through transfection of sensitive L1210 cells with a plasmid containing a full-length human *ABCB1* gene [20].

## 2. Results

### 2.1. Characterization of ABC Transporter Expression in Parental Cell Lines and Their Resistant Variants

The parental cell lines (L1210, SKM-1, and MOLM-13) show almost unmeasurable levels of P-gp gene expression on mRNA levels. In contrast, human *ABCB1* gene (h) expression in cell variants L1210/T, MOLM-13/vcr, and SKM-1/vcr is significant. Alternatively, the L1210/R cell variant expresses the murine *ABCB1* gene (m). The expression of MRP1 remains constant in both sensitive and resistant cells, and it remains unaffected even after a 24 hour-long incubation with inhibitors (TQR or ZSQ). The only exception to this are cells pre-incubated with DMSO, used as a control experiment, where the level of MRP1 is slightly increased (Figure 1)**.** In all four resistant cell variants, P-gp levels are also detectable by Western blotting, as we have described in several of our papers [19,20,21]. We continuously monitored the expression levels of this gene at mRNA and protein levels during ongoing experiments.

### 2.2. Effect of ZSQ and TQR on Leukemic Cells Overexpressing P-Glycoprotein

For intracellular Ca^2+^ concentration measurements, cells must be intact and without signs of cell damage. Therefore, a safe concentration of TQR or ZSQ that does not induce cell damage must be determined. Thus, we established a concentration range where neither TQR nor ZSQ significantly damaged the cells. The most convenient method for determining cellular damage is double staining with annexin-V-labeled fluorescein isothiocyanate (FAV) and propidium iodide (PI), which provides data on the onset of either apoptosis (cells stained by FAV) or necrosis (cells stained by PI), while also identifying viable cells (unstained cells) as well as cells in the late phase of cell death (labeled by both FAV and PI). Incubating the cells in media containing 0.05 to 5.00 μM TQR or ZSQ can be considered safe, as a predominant portion of the cells remain viable when cytometrically detected with FAV and PI (Figure 2 and Figure 3). Under these inhibitor concentrations, cell viability was 80%, except for L1210 and L1210/T cells incubated in the presence of 50 μM ZSQ, where only two-thirds of the cells were viable (Figure 2). The cell viability was measured after a 45 min incubation period for all examined inhibitor concentrations. Regardless of the level of P-gp expression in the cells, there was no observed correlation between the concentration of TQR or ZSQ and a decrease in cell viability. Among all cell lines and their P-gp positive variants, cells entering necrosis (labeled by PI) predominated, while the proportion of apoptotic cells (labeled by FAV), as well as cells in the late phase of cell death (cells labeled by both FAV and PI), was lower (Figure 2 and Figure 3).

### 2.3. Evaluation of the Inhibitory Effects of Zosuquidar and Tariquidar

Flow cytometric assessment of calcein retention is a valuable tool for assessing P-glycoprotein transport activity. Initially, non-fluorescent calcein/AM diffuses across the cell membrane but is immediately effluxed in P-gp-positive cells, whereas free calcein is retained in P-gp-positive cells [22] as it is not a substrate of P-gp [23]. Free calcein also exhibits prominent fluorescence upon de-esterification [24]. This retention discrepancy is crucial to understanding P-gp function and can be modulated by P-gp inhibitors. Because of these properties, calcein/AM is often used as a model substrate for P-gp studies [25].

In our previous work, we demonstrated the ability of TQR to increase calcein retention in both P-gp positive variants of L1210 cells [20]. Figure 4 and Figure 5 show the effect of ZSQ at concentrations of 0.25 and 0.50 μM on calcein fluorescence retention in P-gp negative parental cells and their P-gp positive variants, respectively. Both concentrations were optimized in a set of preliminary experiments, to achieve maximum efficiency (not shown). Quantification of the data revealed statistical significance for the T and SKM-1/vcr variants (Table 1).

The fluorescence shift was not as markedly apparent in SKM-1/vcr and MOLM-13/vcr variants (Figure 4) as in P-gp positive R cells of the L1210 cell line (Figure 3). Moreover, T-variants suggested a similar behavior as in human AML variants (SKM-1/vcr and MOLM-13/vcr). Human P-gp expressed in the T-variant of mouse L1210 cells is probably responsible for this similarity. However, ZSQ successfully showed its ability to reverse P-gp-mediated vincristine resistance. Both studied concentrations provided complete (R, T, SKM-1/vcr) or at least partial (MOLM-13/vcr) restoration of calcein fluorescence.

### 2.4. Flow Cytometric Evaluation of Intracellular Calcium Levels

A flow cytometer was used to determine intracellular calcium levels with Fluo-3/AM in a single living cell in real time. The method used in this section was based on a paper written by Schepers et al. in 2009 [9]. Cells were suspended in a HEPES-buffered calcium-free medium and, prior to each measurement, extracellular calcium was added in the form of CaCl_2_. A cutoff value of 210 s was used for each measurement of parallel samples (*I1*–*I5*). Subsequently, cells were loaded with Fluo-3/AM in the presence or absence of either TQR or ZSQ (45 min incubation period with the inhibitors). The fluorescence signal of the Fluo-3/Ca^2+^ complex in single cells was measured and quantified using the FL1-A channel. The logarithm of the ratios between cell fluorescence intensities in the presence or absence of Fluo-3/AM was used as a measure of intracellular Ca^2+^. Control samples without the addition of Fluo-3/AM served as a blank. Upon excitation at 488 nm, the Fluo-3/Ca^2+^ complex emits photons with a typical wavelength of 525 nm [26]. Satisfactory cell fluorescence was observed only in cell variants not expressing P-gp. Conversely, fluorescence was relatively low in cells with positive P-gp expression. This phenomenon is due to the barrier against cell filling with Fluo-3/AM, presented by the efflux activity of P-gp, as Fluo-3/AM is one of its substrates [16]. Figure 6A,B shows dot plots of continuous calcium dynamics in each sample after treatment with different inhibitors. After the addition of extracellular calcium to the cells, there was an immediate increase in the signal. In samples with inhibited P-gp, the signal intensity was at the same level as in cells not expressing P-gp. Consistent with this, both P-gp inhibitors (TQR and ZSQ) antagonized the efflux of Fluo-3/AM and allowed its intracellular loading, de-esterification, and subsequent formation of the Fluo-3/Ca^2+^ complex. As a result, the fluorescence intensity closely matched that observed in the sensitive cell variants (Figure 6C,D).

### 2.5. Observation of Intracellular Fluo-3/Ca^2+^ Complex in Confocal Microscopy

Confocal microscopy provides a visual approach to the monitoring of intracellular Ca^2+^ transport. As a follow-up to a study from Sulova et al. (2005, 2009) [18,27], the interplay between P-gp-mediated MDR and intracellular calcium homeostasis was evaluated. Differences in terms of extracellular calcium loading and subcellular compartmentalization of Ca^2+^ were observed.

Prior to the measurement, the cells were placed in a calcium-free medium (see Materials and Methods—Section 4.4), as calcium in commercially available media would interfere with the measurements. Regardless of the presence of Fluo-3, P-gp positive variants showed no green fluorescence. As in previous cases, the Ca^2+^ indicator presumably underwent massive efflux as a P-gp substrate. However, after the addition of 0.25 μM ZSQ, the green fluorescence was reliably restored. Hoechst 33,342 was used to visualize the nuclei (Figure 7).

## 3. Discussion

The fate of cells largely depends on proper regulation of free calcium in the cytosol. Therefore, precise regulation of the calcium signal must be maintained in cells for proper cell function. While the elevation of intracellular Ca^2+^ levels to toxic levels kills non-cancerous cells, cancer cells are less sensitive to this alteration as deregulation of cytosolic free calcium levels occurs more frequently. This phenomenon is also considered a hallmark of cancer cells [28]. Unregulated Ca^2+^ homeostasis is exploited by cancer cells to avoid apoptotic cell death pathways [2,29]. Alterations in calcium signaling are also frequently observed and associated with epithelial-mesenchymal transition (EMT), and other metastatic invasion of various cancers. In addition, an association between the induction of EMT and increased expression of specific members of the ABC transporter superfamily has been previously reported [30]. Overexpression of P-glycoprotein (P-gp), a member of the ABC transporter family of ABCB1, is associated with the development of multidrug resistance (MDR), which is the most common cause of treatment failure in neoplastic diseases, including AML [31].

Cells without significant overexpression of P-gp or other membrane transporters capable of expelling intracellular calcium indicators (such as Fluo-3/AM) can be used for sensitive monitoring of intracellular calcium levels. However, in cells overexpressing P-gp, these efflux pumps protect the extracellular space from filling with indicators [25] and make the measurement of calcium levels impossible [17,27]. Therefore, to make calcium levels measurable, it is necessary to use a suitable P-gp efflux pump inhibitor.

In this study, two P-gp inhibitors Zosuquidar (ZSQ) and Tariquidar (TQR) were evaluated. Both were used to enable the detection of cytosolic calcium levels in P-gp positive leukemia cell variants (R, T, SKM-1/vcr, and MOLM-13/vcr), and the results obtained were compared with those from P-gp negative counterparts (S, SKM-1, and MOLM-13). Inhibitor concentrations ranged from 0.05 to 5.00 μM, which we predicted to have little effect on cell viability. When both inhibitors were used in this concentration range, a predominant fraction of cells were viable. However, the addition of ZSQ to a final concentration of 5.00 μM induced a 2.5-fold increase in apoptosis in S and T cells (Figure 2). We observed no comparable increase in apoptosis in either the sensitive or resistant variants of SKM-1 and MOLM-13 cells. Conversely, none of the concentrations tested resulted in a significant decrease in viability (see Figure 3). These results were further complemented by calcein retention assays to detect the P-gp transport function. For Tariquidar (at a concentration of 500 nM), we have documented its ability to retain calcein in previous work [20].

Although both TQR and ZSQ are third-generation P-gp inhibitors [32], there are fundamental differences between them in terms of their mechanism of action. TQR is considered a highly specific, non-competitive P-gp inhibitor [33]. The P-gp reversal potency of TQR is achieved by blocking the function of P-gp. This effect may be derived either from substrate binding inhibition (due to occlusion of the binding site), ATP hydrolysis prevention (due to the cessation of the P-gp reaction cycle), or a combination of both [34]. In contrast, ZSQ, as a competitive inhibitor of P-gp, directly occupies the substrate-binding region and prevents active transport [35]. The nature of the competitive inhibition implies that an increased concentration of the drug as a substrate decreases the binding of Zosuquidar. However, despite this difference, both Zosuquidar and Tariquidar can restore calcein retention in cells at submicromolar concentrations, and 500 nM of one or the other is sufficient to completely block P-gp transport and restore calcein retention [20] (Figure 4 and Figure 5, and Table 1).

Using Fluo-3/AM initially seemed like a relatively non-invasive way of intracellular calcium screening in living cells, but the method soon faced severe limitations due to massive P-gp-mediated extrusion [36]. First-generation P-gp inhibitors—cyclosporine A and verapamil—were tested in the past as an attempt to overcome this issue. There are published data in P-gp positive cells that verapamil will ensure retention of Calcein-AM in calcein retention assays [37]; however, in combination with Fluo-3/AM it may fail to ensure its full retention [17]. In our previous work, we were able to restore JC-1 retention in R and T cells with Tariquidar but not with cyclosporine A or verapamil [20]. Verapamil, as a blocker of voltage-dependent calcium channels, may affect intracellular Ca^2+^ homeostasis and could interfere with the measurement of free Ca^2+^ levels by Fluo-3 [17,38]. Based on these facts, we have decided to test TQR and ZSQ, as these are not expected to directly affect the Ca^2+^ homeostasis. Both P-gp inhibitors managed to restore the Fluo-3 fluorescence in P-gp positive cell variants with a visible shift in the fluorescence intensity (Figure 6A (R, T) and Figure 6B (SKM-1/vcr and MOLM-13/vcr)). Two final concentrations of ZSQ (0.25 and 0.50 μM) and a final concentration of 0.50 μM TQR were tested. Each of the studied concentrations caused an increase in the fluorescence of P-gp positive cell variants up to a 3-fold increase (R variant of L1210 cells), with statistically significant results obtained in each case. When added to P-gp positive L1210 cell variants, the final concentration of 0.50 μM ZSQ seemed to operate slightly more efficiently on the restoration of the Fluo-3 fluorescence than the same amount of TQR (Figure 6C).

It was previously emphasized by Hagen et al. (2012) that the behavior of calcium-sensing molecules can dramatically change when loaded into living cells [39]. This statement especially applies to cancer cell models with acquired MDR. Therefore, this paper aimed to study the differences in the intracellular compartmentalization of the fluorescent signal using high resolution confocal microscopy images. P-gp negative cells failed to be properly loaded with Fluo-3/AM, visible as an absence of green fluorescence (Figure 7). However, consistent with our previous results, both studied concentrations of ZSQ (0.25 and 0.50 μM) could reliably restore the missing fluorescence. A correlation was hard to find while visually comparing murine L1210 and human SKM-1 and MOLM-13 cells. In most cases, intracellular calcium was found to be localized scattered within the cytoplasm, while evenly coating the cells. On the other hand, in the case of sensitive SKM-1 and MOLM-13 cells, calcium-rich regions were predominantly found forming clumps along the cell surface coat. A similar view was observed in T cells after the addition of 0.50 μM ZSQ, which is another sign of the correlation between human AML cells and the T variant of L1210 cells. However, human AML cell models showed an overall lower Fluo-3 fluorescence intensity when compared with murine ALL variants. This pattern, however, seemed to be reversible with the addition of ZSQ. An increase in the calcium signal was observed in both MOLM-13 variants as an amplified circle of green fluorescence along the nuclear envelope.

Using Fluo-3/AM as a fluorescent calcium sensing probe can pose major obstacles for various reasons explained in this paper. Nevertheless, we can conclude that both Tariquidar and Zosuquidar showed their ability to inhibit P-glycoprotein activity in resistant cells. Moreover, the results of our study show the applicability of these inhibitors in functional calcium kinetics evaluation.

## 4. Materials and Methods

### 4.1. Cell Lines

Mouse acute lymphocytic leukemia cells: L1210 (ACC 123) and human acute myeloid leukemia cell lines: SKM-1 (ACC 547) and MOLM-13 (ACC 554) were obtained from the Leibniz-Institut DSMZ-Deutsche Sammlung von Mikroorganismen und Zellkulturen GmbH (Braunschweig, Germany). P-gp positive L1210 cell variants were derived from parental, sensitive (S) cells either by vincristine selection (R) or stable transfection with the P-gp gene from the Addgene plasmid 10957 (T) as described by Elefantova et al. 2018 [20]. P-gp positive SKM-1/vcr and MOLM-13/vcr variants were obtained by the selection of parental cells and their adaptation to vincristine (VCR). The expression of P-gp in each cell variant was periodically controlled at mRNA levels via RT-PCR.

### 4.2. Apoptosis and Necrosis Detection Assay

Cells (5 × 10^5^/1 mL) were cultured with ZSQ (Merck LifeSciences, Rahway, NJ, USA) or TQR (SelleckChem, Houston, TX, USA)—final concentrations: 0.05 μM, 0.1 μM, 0.25 μM, 0.50 μM, and 5.00 μM, which were selected according to preliminary cytotoxicity assays. The incubation period lasted for 45 min at 37 °C in the dark. Further, the cells were centrifuged for 5 min at 2500 rpm and resuspended in binding buffer (10 mM HEPES (Merck LifeSciences, Rahway, NJ, USA)/NaOH, pH 7.4, 140 mM NaCl, 5 mM CaCl_2_) (all purchased from Mikrochem, Pezinok, Slovakia). The addition of Annexin V FLUOS (A-V) (final concentration 0.5 μg/mL) (Roche Diagnostics GmbH, Mannheim, Germany) was followed by a 15-min incubation period in the dark at room temperature. Before each measurement, propidium iodide (PI) (final concentration 10 μg/mL) (Merck in Slovakia, Bratislava, Slovakia) was added to each sample. Samples were analyzed by flow cytometry using the Accuri C6 flow cytometer (BD Bioscience, San Jose, CA, USA).

### 4.3. Calcein/AM Assay

We evaluated the transport activity of P-gp by calcein/AM assay. Cells (5 × 10^5^/1 mL) were incubated in a phenol-free RPMI-1640 medium (Biosera, Kansas City, MO, USA) with the addition of ZSQ in a final concentration of 0.25 and 0.50 μM for 45 min at 37 °C in the dark. Afterwards, Calcein-AM (final concentration of 0.01 µM) (Merck LifeSciences, Rahway, NJ, USA) was added to the samples, and the incubation continued for another 10 min (L1210 cell variants)/5 min (SKM-1 and MOLM-13 cell variants). Subsequently, the cells were centrifuged and washed with phenol-free RPMI-1640 medium (Biosera, Kansas City, MO, USA). PI (final concentration 10 μg/mL) was added to each sample before the analysis. The assay was performed on the Accuri C6 flow cytometer.

### 4.4. Evaluation of Cytoplasmic Calcium Kinetics

Cells (5 × 10^5^/500 μL) were incubated in a calcium-free culture medium. The medium was prepared according to the article of Sulova et al. 2009 [27]. Our medium contained 10 mM HEPES (Merck LifeSciences, Rahway, NJ, USA); 10 mM glucose; 117 mM NaCl; 3 mM KCl, and 1 mM MgSO_4_ (all from Mikrochem, Pezinok, Slovakia). The individual reagents were dissolved in deionized water and, finally, the pH was adjusted to a value close to the physiological pH (pH = 7.4). Next, ZSQ or TQR was added to a final concentration of 0.25 µM and/or 0.50 µM. Finally, Fluo-3/AM solution (44 μL 20% Pluronic/4 μg Fluo-3/AM) was added to a final concentration of 4 μM. The cells were incubated for 60 min at 37 °C in a thermoshaker in the dark. After the end of the incubation period, the cells were washed three times with the calcium-free culture medium. Firstly, we determined the baseline fluorescence, without the additional Ca^2+^ ions. Secondly, external CaCl_2_ was added (final concentration of 1.60 mM) to each sample. Samples were analyzed by flow cytometry using the Accuri C6 flow cytometer.

### 4.5. Detection of Intracellular Calcium Transport

Cells (5 × 10^5^/500 μL) were incubated in a calcium-free culture medium (as described above—4.4). The addition of ZSQ represented 0.25 μM and/or 0.50 μM final concentrations. The Fluo-3/AM solution (44 μL 20% Pluronic/4 μg Fluo-3/AM) (Merck in Slovakia, Bratislava Slovakia/Invitrogen, Carlsbad, CA, USA) was added to a final concentration of 4 μM. Finally, cells were incubated for 30 min at 37 °C in the dark. After the end of the incubation period, the cells were washed 3 times with calcium-free culture medium. A fluorescent probe, Hoechst 33,342 (final concentration 0.4 mM) (Merck LifeSciences, Rahway, NJ, USA), was added to each sample with an additional incubation period of 5 min. As the last step of sample preparation, external CaCl_2_ was added to each sample to a final concentration of 1.60 mM. Visualization was performed on a TCS SP2 AOBS laser confocal system (Leica Microsystems, Wetzlar, Germany) with a DMIRE-2 inverted microscope (Leica, Wetzlar, Germany).

### 4.6. Quantitative PCR

Cells (2 × 10^6^/4 mL) were cultured with ZSQ (Merck LifeSciences, Rahway, NJ, USA) or TQR (SelleckChem, Houston, TX, USA)—final concentrations of 0.25 μM and 0.50 μM, and 500 nM—for 24 h at 37 °C in a 5% CO_2_ atmosphere.

Total RNA was isolated from using the GenElute™ Mammalian Total RNA Miniprep Kit (Thermo Fisher Scientific, Waltham, MA, USA) according to the manufacturer’s instructions. cDNA synthesis was performed using 1 μg of template RNA, Random Hexamer primers (100 pmol), 4 μL of reaction buffer, 0.5 µL RiboLock™ RNase Inhibitor, 2 µL dNTP mix (10 mM each), 1 µL RevertAid H Minus Reverse Transcriptase (all from Thermo Fisher Scientific, Waltham, MAUSA), and DEPC-treated water at a total volume of 20 µL.

The qPCR was run on a 96-well microtitration plate using a CFX Connect real-time System (Bio-Rad, Hercules, CA, USA), in a 10 μL solution containing 500 ng of cDNA, 5 μL of 2x iTaq Universal SYBR^®^ Green Supermix (Bio-Rad, Hercules, CA, USA), 0.7 μL of primer solution at a concentration of 5 μmol/L, and 2.6 μL of RNase-free UltraPureTM DEPC-treated water. The thermal cycling conditions were as follows: polymerase activation (95 °C, 3 min); PCR—39 cycles: denaturation (95 °C, 30 s), and annealing/extension (60 °C; 1 min). The sequences of the primers that were used are summarized in Table 2. The samples were measured in triplicates. The relative expression of individual genes was evaluated by the Livak method (2^−ΔΔCT^).

### 4.7. Statistical Analysis and Data Processing

Numerical data are expressed as mean values ± SD of at least three independent measurements. Statistical significance was assessed using an unpaired Student’s *t*-test using SigmaPlot 8.0 software (Systat Software, Inc., San Jose, CA, USA). A *p*-value of less than 0.05 was considered significant.

## 5. Conclusions

Intracellular Ca^2+^ as a secondary messenger plays a role in healthy cellular homeostasis and its alterations are responsible for the development of various pathological processes. Deregulated Ca^2+^ signaling is associated with multilevel manifestation of typical cancer features [28]. Alterations in intracellular Ca^2+^ homeostasis are also expected in the development of drug resistance [40]. As a very common case associated with P-gp overexpression, an interplay between multidrug resistance and calcium signaling was highlighted [27]. Therefore, changes in intracellular Ca^2+^ levels need to be monitored concerning P-gp expression and the development of drug resistance. As an efflux pump, P-gp can eliminate the majority of available intracellular calcium indicators (e.g., Fluo-3/AM or Fura-2/AM) from the intracellular compartment. Therefore, monitoring intracellular Ca^2+^ concentration is obstructive. In the current paper, we developed a procedure for reliable measurement of intracellular Ca^2+^ levels by monitoring Fluo-3 fluorescence in single cells using fluorescent flow cytometry. Stable retention of Fluo-3 in P-glycoprotein positive cells is achieved by using two third-generation P-gp inhibitors, Zosuquidar or Tariquidar, at their effective concentration of 500 nM.

## Figures and Tables

**Figure 1 ijms-25-03107-f001:**
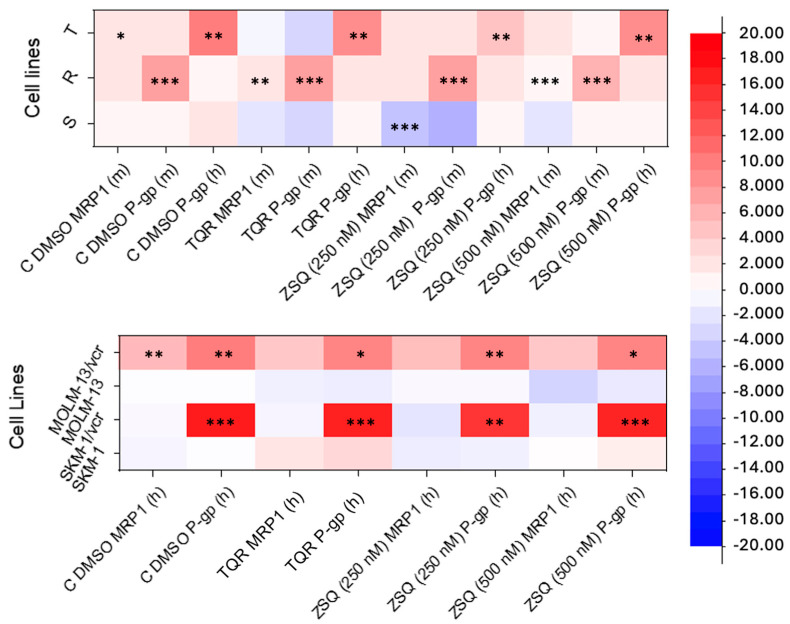
Heat maps representing quantitative PCRs after a 24 hour-long incubation period with ZSQ or TQR. Neither of the inhibitors affected P-gp expression. Primers used against mouse genes are indicated as ‘m’, while primers used against human genes are indicated as ‘h’. The data were normalized to β-actin expression and are shown as the mean of three independent measurements. Significant differences in data were observed compared to the values obtained in sensitive cells. Statistical significance: * *p* ≤ 0.05, ** *p* ≤ 0.01, *** *p* ≤ 0.001.

**Figure 2 ijms-25-03107-f002:**
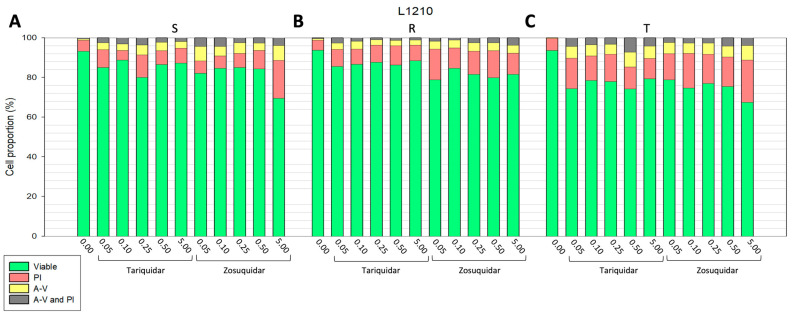
Flow cytometric evaluation of the effect of ZSQ or TQR on P-gp negative S (**A**), P-gp positive R (**B**), and T (**C**) L1210 cells. Samples were incubated with or without inhibitors for 45 min. Fluorescent staining was assessed using Annexin V FLUOS (A-V; yellow; apoptotic cells) or propidium iodide (PI; red; necrotic cells). Unstained cells are considered viable (green), and double-stained cells represent a subpopulation of late apoptotic/necrotic cells (A-V and PI; gray). Values illustrated on the *x*-axis are micromolar concentrations of inhibitors. Our data represent mean values from three independent measurements.

**Figure 3 ijms-25-03107-f003:**
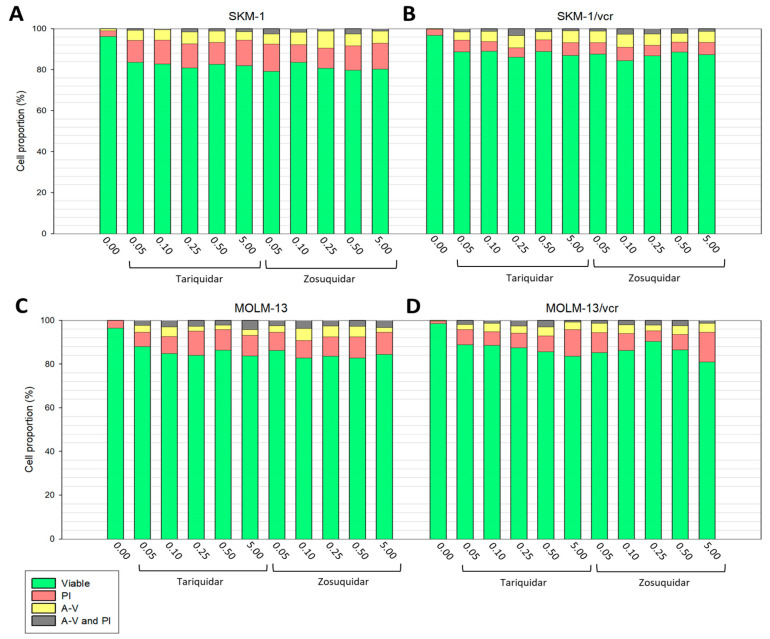
Flow cytometric evaluation of the effect of ZSQ or TQR on sensitive or resistant variants of SKM-1 (**A**,**B**) and MOLM-13 cells (**C**,**D**). For further information, see the description of Figure 2.

**Figure 4 ijms-25-03107-f004:**
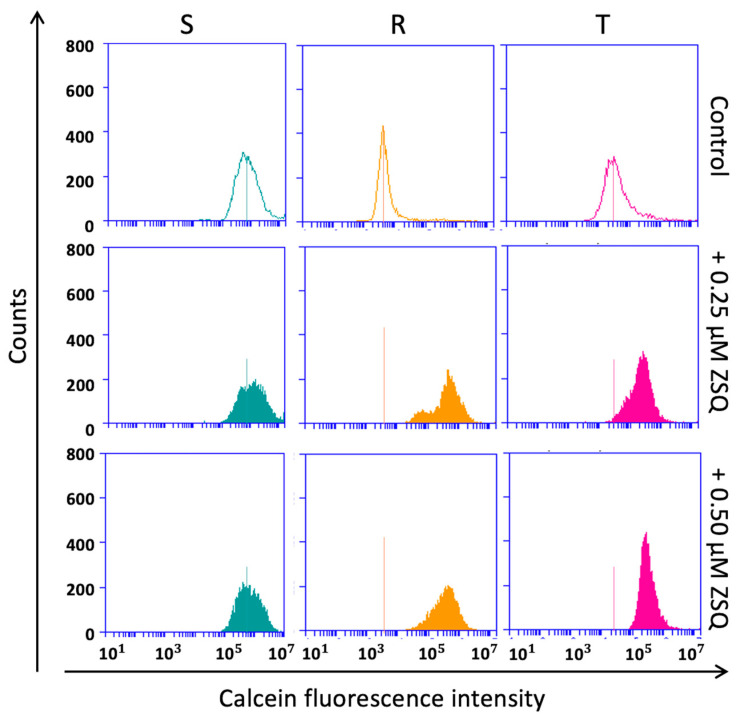
Flow cytometric calcein retention assay on three variants of L1210 cells. The test was performed on a flow cytometer with a cutoff value of 10,000 events. The transport activity of P-gp was monitored using the fluorescent dye Calcein-AM, which was added to a final concentration of 0.1 µM. We detected a visible effect of ZSQ in all P-gp positive cell variants. Tests with P-gp negative variants served as control groups. This figure shows a selection of representative histograms from at least three independent measurements.

**Figure 5 ijms-25-03107-f005:**
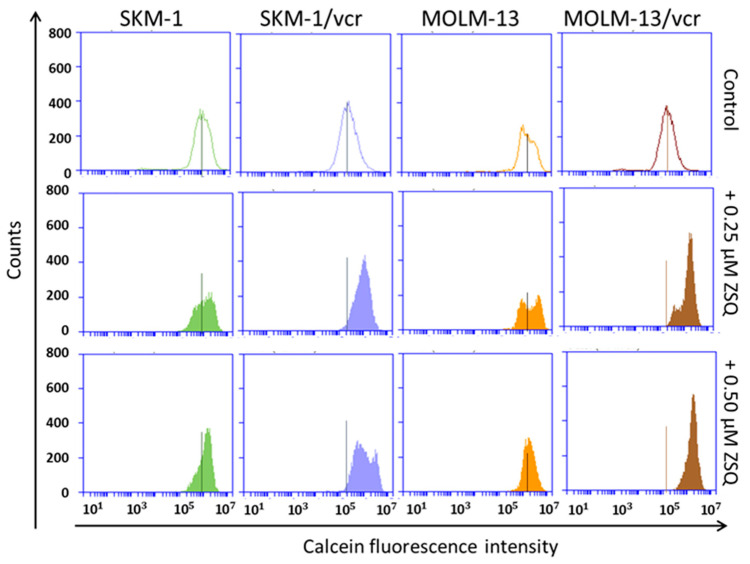
The flow cytometric Calcein-AM efflux assay was performed on sensitive and resistant variants of SKM-1 and MOLM-13 cells. According to our pilot experiments, the cultivation period of Calcein-AM with our samples was shortened to 5 min. However, the final volume of the fluorescent dye added to the samples remained unchanged. The figure serves as an illustration of representative histograms from at least three independent measurements.

**Figure 6 ijms-25-03107-f006:**
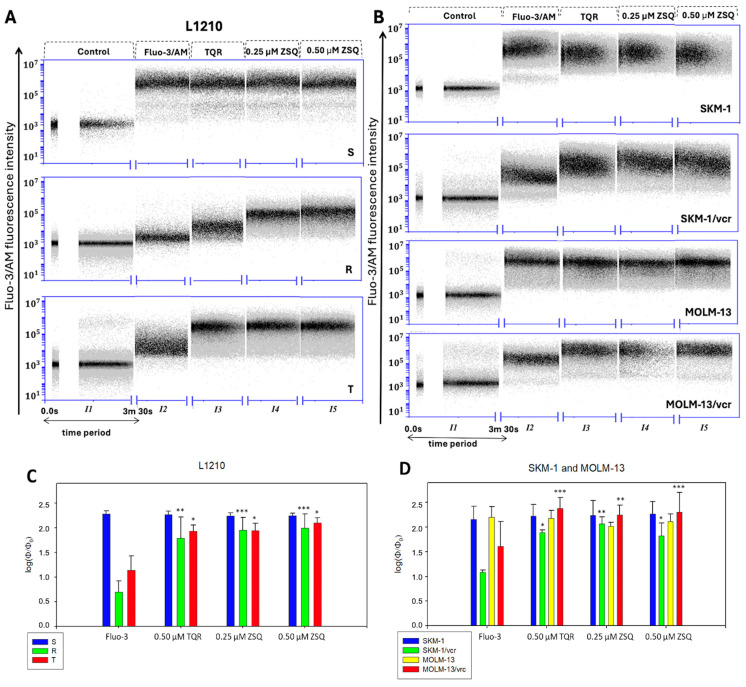
(**A**,**B**) Fluo-3 fluorescence plots were obtained by flow cytometry. Before each measurement, samples were stimulated by adding CaCl_2_, which represented extracellular Ca^2+^. Control samples represent completely unstained cells, emitting a baseline fluorescence. Each time period of 210 s represents an interval labelled from *I1* to *I5* to distinguish parallel samples. Fluo-3 fluorescence was detected using an argon laser of the Accuri BD C6 flow cytometer. Further quantification of these data allowed us to analyze the obtained signal. (**C**,**D**) Quantification of the flow cytometric calcium flux assay using Fluo-3/AM. Illustrated data were obtained by subtraction of fluorescence intensity—ϕ (median values) and the baseline fluorescence—ϕ_0_ (control samples), all from three independent measurements. With further logarithmization of the obtained data, the proportion of intracellular Ca^2+^ levels was assessed as (log(ϕ/ϕ_0_)) ± SD. A similar level was maintained in the case of all sensitive variants ((**C**)–S, (**D**)–SKM-1, MOLM-13). However, Fluo-3 samples of P-gp positive variants ((**C**)–R, T, (**D**)–SKM-1/vcr, MOLM-13/vcr) showed a drop, due to increased Fluo-3/AM efflux as a P-gp substrate. However, a rise was seen in the case of every P-gp positive sample treated with either TQR or ZSQ. Statistical significance as follows: * *p* ≤ 0.05, ** *p* ≤ 0.01, *** *p* ≤ 0.001.

**Figure 7 ijms-25-03107-f007:**
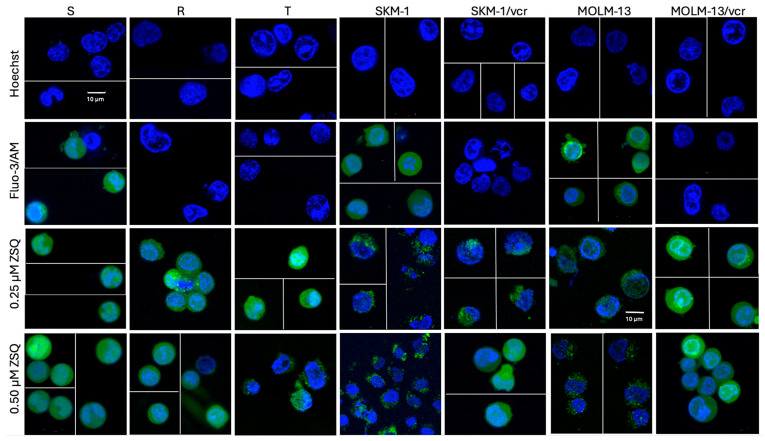
Qualitative determination of calcium ion transport was performed on a TCS SP2 AOBS laser scanning confocal fluorescence system with a DMIRE2 inverted microscope from Leica Microsystems (Wetzlar, Germany). The fluorescent probe Hoechst 33342 (blue), which binds to the A-T regions of DNA, was used to visualize the nuclei. After the addition of 0.25 μM ZSQ to the resistant variants, we observed a restoration of the green fluorescence represented by Fluo-3 (green). This fact proves the inhibition of the efflux activity of P-gp. The figure shows a collage of representative pictures (scale bar: 10 μm) displaying multiple cells from at least three independent measurements.

**Table 1 ijms-25-03107-t001:** Quantification of the flow cytometric Calcein-AM assay data.

Cell Variant		Control (Unstained)	Control (Calcein-AM)	ZSQ (0.25 μM)	ZSQ (0.50 μM)
S	means ± SD *p*-value	3.36 ± 0.02	6.04 ± 0.19 0.027 *	6.31 ± 0.30 0.126	6.31 ± 0.35 0.079
R	means ± SD *p*-value	3.34 ± 0.01	4.51 ± 0.16 0.029 *	5.91 ± 0.24 0.056	5.85 ± 0.29 0.102
T	means ± SD *p*-value	3.22 ± 0.01	5.32 ± 0.20 0.038 *	5.74 ± 0.18 0.037 *	5.88 ± 0.18 0.031 *
SKM-1	means ± SD *p*-value	3,26 ± 0,03	5.95 ± 0.15 0.004 **	5.85 ± 1.09 0.060	5.98 ± 0.14 0.002 **
SKM-1/vcr	means ± SD *p*-value	3.28 ± 0.01	5.35 ± 0.20 0.020 *	6.16 ± 0.12 0.001 ***	6.18 ± 0.01 0.001 ***
MOLM-13	means ± SD *p*-value	3.36 ± 0.04	6.04 ± 0.22 0.028 *	6.09 ± 0.30 0.072	6.06 ± 0.20 0.022
MOLM-13/vcr	means ± SD *p*-value	3.36 ± 0.15	5.16 ± 0.34 0.067	5.83 ± 0.42 0.111	5.86 ± 0.36 0.128

Individual values were obtained from at least three independent measurements. Data were quantified using mean values, with further logarithmic modification. Only viable (PI-negative) cells were considered. Standard deviation was used to determine measurement errors. *p*-values were calculated using a paired Student’s *t*-test. Statistical significance as follows: * *p* ≤ 0.05, ** *p* ≤ 0.01, *** *p* ≤ 0.001.

**Table 2 ijms-25-03107-t002:** Primer sequences.

Gene	Forward Primer	Reverse Primer	Size (bp)
Human *GAPDH*	ATCGTGGAAGGACTCATGACC	GCCATCACGCCACAGTTTC	90
Mouse *GAPDH*	AGCTTCGGCACATATTTCATCTG	CGTTCACTCCCATGACAAACA	89
Human *ABCB1*	GACAGCTACAGCACGGAAGG	CTGAAGCACTGGGATGTCCG	108
Mouse *Abcb1*	GGCTGTTAAAGGTAACTCC	TGTTCTCTTATGAATCACGTA	152
Human *MRP1*	CCGTGTACTCCAACGCTGACAT	ATGCTGTGCGTGACCAAGATCC	145
Mouse *MRP1*	ACCAGCAACCCCGACTTTAC	TGGTTTTGTTGAGGTGTGTCA	151

## Data Availability

Data is contained within the article.

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
