# Peer review of "Zosuquidar: An Effective Molecule for Intracellular Ca2+ Measurement in P-gp Positive Cells"

_ijms, 2024, doi:10.3390/ijms25063107_

Round 1

Reviewer 1 Report

Comments and Suggestions for Authors

The manuscript provides highly valuable methodological approach capable to monitor intracellular calcium levels in cellular in vitro models with overexpression of multidrug resistence components (ABC transporters). The major focus of this mansucript is on P-gp and authors showed that the calcein retention can be modified efectively using nanomolar concentrations of P-gp inhibitors. I can recommend to accept the manuscript after minor revision (see below).

Specific comments:

1. Results, chapter 2.1, line 89: Here, it would be valuable to report in a more detail the data presented in Figure 1. E.g., the changes in P-gp/ABCB1 should be reported here; the levels of MRP1/ABCC1 are not discussed at all. What mean the letters “m“ and “h“ in brackets? How statistical significance was calculated when controls (DMSO treated cells) show also significant changes?

2. Results, lines 106-109: I cannot see the viability data in Figure 1. The of chapter 2.2 should be focused on Figures 2 and 3.

3. Figure 2: the L1210 cell variants should be marked with the letters S, R or T.

4. Legend to Figure 3: For further information see the description of Figure 2.

5. Chapter 2.3, lines 132-144: Authors here repeat the basic knowledge on the effect of P-gp level on calcein retention. This paragraph could be shortened.

6. Chapter 2.3, line 146: The data presented in Figures 4 and 5 are reported here.

7. Figures 6A-B are not reported; the from Figures 6C-D) are reported as Figures 5C-D), see line 200.

8. Chapter 2.5, lines 223-233: Figure 7 is not mentioned in this part of results.

9. Discussion, line 277: The data presented in other cell lines (Figure 3) should be also dicussed here.

10. Discussion, lines 331-332: an ability of both chemical inhibitors to reverse P-gp-mediated vincristine resistance is not presented in this paper. The authors should either show the data or to use suitable reference.

Comments on the Quality of English Language

Some minor editing could improve the quality of English.

Author Response

Thank you for your valuable feedback.

Reviewer 2 Report

Comments and Suggestions for Authors

The authors suggest measuring intracellular calcium ion levels using the FURA-3 indicator and two third-generation P-gp inhibitors. I like the idea, although I don't see any new content, and I have serious concerns about the implementation:

1. The P-gp inhibition presented in the article is not perfect. The broad dye uptake (spanning 2 orders of magnitude) appearing in the FACS diagram, the inhibition with two peaks, can be attributed to the wrongly adjusted cell number/dye concentration/inhibitor ratio.

2. The function of P-gp is presented with the help of Calcein-AM, which is also a dye used to detect a calcium ion. In my opinion, it would have been worthwhile to use a different type of dye here, such as Rhodamine 123 dye, which glows green like Calcein-AM.

3. In the P-gp mRNA expression test, MRP1 appears significantly in the case of some cell lines/treatments. Both Calcein-AM and Fura-3 are also MRP1 substrates. The authors do not attempt to inhibit this transporter, so we do not know for sure what we are seeing.

4. It would have been useful if P-gp expression was shown not only at the mRNA level, but also with actual antibody labeling and protein expression.

5. Interestingly, the L1210 basal cell shows high MDR activity factor (MAF). The drug-treated cell gives a higher MAF than the transfected one (calculated from the data in the article).

5. In the case of SKM-1 cells, the smallest ZSQ concentration kills the cells the best, which also affects calcium homeostasis. The same problem appears in Fig. 2 B of the L1210 cell, which is presumably the R cell line.

6. I wouldn't call Figure 4 representative, since there are serious shifts where there shouldn't be.

7. In the case of Fig. 6, it can be seen in several places that the cell concentration was used decreased during the 3 min 30 sec measurement, and in several cases the fluorescence intensity is not stable either. How can you explain that, for example, the L1210R is charged lower with TQR than with ZSQ? It should be the other way around.

8. Neither the number of cells nor the appearance/magnitude of fluorescence in the microscopic image is the same as in the control?

9. Calcein ion chelator would have been useful, e.g. also present EDTA measurements.

Comments on the Quality of English Language

The English language of the manuscript is good and understandable. The numbering of the pictures is wrong.

Author Response

Thank You for your valuable feedback

Round 2

Reviewer 2 Report

Comments and Suggestions for Authors

I accept this answer.

Comments on the Quality of English Language

No comments.